
# Entanglement in the full state vector of boson sampling

Yulong Qiao[1,2], Joonsuk Huh[3,4,5★] and Frank Grossmann[1†]

**1** Institut für Theoretische Physik, Technische Universität Dresden, 01062 Dresden, Germany
**2** Max-Planck-Institut für Physik komplexer Systeme,
Nöthnitzer Str. 38, 01187 Dresden, Germany
**3** Department of Chemistry, Sungkyunkwan University, Suwon 16419, Korea
**4** SKKU Advanced Institute of Nanotechnology (SAINT),
Sungkyunkwan University, Suwon 16419, Korea
**5** Institute of Quantum Biophysics, Sungkyunkwan University,
Suwon 16419, Republic of Korea

★ joonsukhuh@gmail.com , † frank.grossmann1@tu-dresden.de

## Abstract

The full state vector of boson sampling is generated by passing $S$ single photons through beam splitters of $M$ modes. We express the initial Fock state in terms of $2^{S-1}$ generalized coherent states, making possible the exact application of the unitary evolution. Due to the favorable polynomial scaling of numerical effort in $M$, we can investigate Rényi entanglement entropies for moderate particle and huge mode numbers. We find symmetric Page curves with a maximum entropy at equal partition, which is almost independent on Rényi index. Furthermore, the maximum entropy as a function of mode index saturates for $M \geq S^2$ in the collision-free subspace case. The asymptotic value of the entropy increases linearly with $S$. In addition, we show that the build-up of the entanglement leads to a cusp in the asymmetric entanglement curve. Maximum entanglement is reached well before the mode population is distributed over the whole system.



# 1  Introduction

As a nonuniversal model of quantum computation based on a linear optical setup, boson sampling has come to the limelight recently [1,2]. In its standard (Fock state) version, a relatively small number $S$ of identical photons is prepared in $M$ (usually on the order of $S^2$) optical modes and is scattered by an interferometer consisting of beam splitters and phase shifters. These passive linear optical elements determine the Hamiltonian, under which the input state evolves. A uniformly ('Haar') random unitary matrix describes the transformation from the input to the output ports [3,4]. As an output it generates single photon states to be measured with simple bucket detectors [2,5], because the probability that two photons arrive at the same detector is considerably small under the so-called collision-free subspace condition. Recently, Gaussian boson sampling has matured into a prime candidate to prove the principle of the advantage (also referred to as "supremacy") of quantum computation over classical computation [6]. A review of the many different variants of boson sampling and their experimental realization as well as its validation, including references to pioneering studies, is given in [5].

Entanglement, as first formulated by Schrödinger in 1935 [7], is one of the most striking properties of quantum systems that it is at the heart of quantum information, quantum computation, and quantum cryptography [8]. Entanglement is a fundamental resource for performing tasks such as teleportation, key distribution, quantum search, and many others; therefore, the calculation of entanglement and related measures [9] in lattice Hamiltonians has gained considerable interest [10–14]. Buonsante and Vezzani, e.g., have performed entanglement studies to investigate quantum phase transitions in the Bose-Hubbard model [10]. In [15], it has been pointed out that even for noninteracting particles, bosonic entanglement creation outmatches the fermionic case due to the larger Hilbert space in the former case, see also [16]. A commonly used measure for estimating the entanglement is given by the Rényi entropy in a bipartite many-body setup [17]. Understanding the build-up of entanglement, e.g., after a quench (a sudden change of the underlying Hamiltonian) is a central focus of the field [18], and an efficient simulation of quench dynamics on a classical computer would be a valuable resource [19]. The exact numerical application of the unitary evolution for lattice systems with more than a handful of particles as well as modes seems elusive, however, due to the growth of the entanglement. In lattice systems relevant for solid-state physics, e.g., matrix product state (MPS) calculations are mainly favorable for area law scaling of the entanglement growth [20]. Although the output state of boson sampling does not follow an area law [13], in [11], the usage of MPS has been advocated for boson sampling (as well as for a fermionic circuit). Therein, results for moderate particle and small mode numbers have been presented. The efficiency of these numerical calculations relies on the restriction to a small bond dimension, which, however, is conflicting with the rapid growth of entanglement.

In the present contribution, we will focus on the entanglement creation in a boson sampling setup and will use generalized coherent states (GCS), also known as SU($M$) coherent states, as basis functions, in order to cope with the implementation of the (time) evolution. In [21], it has been shown that these non-orthogonal basis sets can deal with unitary evolution for exceedingly large Hilbert space dimensions of the Bose-Hubbard model Hamiltonian with onsite interaction and nearest neighbor tunneling. Herein, the case of a unitarily rotated initial state in a boson sampling setup, which is not restricted to nearest neighbor coupling only (but without onsite interaction), will be shown to be another favorable case for the application of GCS. Along the way, we rederive Glynn's formula [22] for the permanent of a square matrix (see Appendix A) by focussing on singly occupied modes with filling factor one. Expanding this state as well as more general states, with a filling factor smaller than one, in terms of GCS is performed *exactly* analytically, by applying Kan's monomial expansion formula [23] to the product of creation operators that appears in the Fock state. After establishing a formula for the trace of powers of the reduced density operator in terms of GCS overlaps only, we study the Rényi entropy for different Rényi index as a function of the subsystem size up to the collision-free subspace limit, where for the total number of modes, it should be sufficient that $M \geq S^2$ [24]. All the results to be presented below are exact and analytical and just require the numerical evaluation of (multiple) finite sums.

The manuscript is structured as follows. In Sec. 2, we briefly review the formalism of boson sampling. Then, in Sec. 3, we lay the foundation for applying the unitary beam splitting operation by using the generalized coherent states as basis functions. To this end, we focus on the exact analytical expansion of the initial Fock state in terms of a finite sum of GCS. We then establish a way to calculate the purity and traces of higher powers of the reduced density operator for computing the Rényi entropy. In Sec. 4, we focus on the creation of entanglement of subsystems due to the random unitary evolution. Though the main focus of our presentation is on the application of the full unitary matrix, we also consider the build-up of entanglement. Sec. 5 concludes the presentation and gives an outlook on future developments. Technical details are given in the appendix.

## 2 Boson Sampling as a unitary evolution

The quantity of interest in theoretical approaches to boson sampling (BS) is the probability $P$ for a given configuration $C$ of $S$ photons distributed over $M$ detectors in the collision-free subspace case, where $M \gg S$ and every mode is maximally singly occupied. This probability is related to the permanent of a submatrix of a (Haar random) unitary matrix with entries $U_{ij}$ [25]. To see this, we briefly review the BS formalism by looking at the full state vector expressed in the Fock state basis $\{|\mathbf{m}\rangle\}$. It is given by [2],

$$|\Psi\rangle = \hat{R}\left(\mathbf{U}^{\mathbf{T}}\right)|\mathbf{n}\rangle = \sum_{C} \gamma_C |\mathbf{m}\rangle, \tag{1}$$

where the evolved initial state $|\mathbf{n}\rangle = |n_1, n_2, \ldots, n_M\rangle$ is expanded in terms of different $|\mathbf{m}\rangle = |m_1, m_2, \ldots, m_M\rangle$. In the collision-free subspace case, the expansion coefficients are given as permanents (per)

$$\gamma_C = \langle \mathbf{m}|\hat{R}\left(\mathbf{U}^{\mathbf{T}}\right)|\mathbf{n}\rangle = \mathrm{per}(U_{\mathbf{nm}}), \tag{2}$$

where $U_{\mathbf{nm}}$ is prepared by taking $n_k$ copies of the $k$-th column and $m_k$ copies of the $k$-th row of the full matrix $\mathbf{U}$. Thus, the probability is identified as $P(C) = |\gamma_C|^2 = |\mathrm{per}(U_{\mathbf{nm}})|^2$. For an instance of three photons, initially in the Fock state $|1, 1, 1, 0, 0, \ldots\rangle$ and finally in

$|1, 0, 1, 0, \ldots, 0, 1\rangle$, the submatrix is constructed as follows:

$$U_{\mathbf{nm}} = \begin{pmatrix} U_{11} & U_{12} & U_{13} \\ U_{31} & U_{32} & U_{33} \\ U_{M1} & U_{M2} & U_{M3} \end{pmatrix}. \tag{3}$$

Furthermore, $\mathrm{per}(\mathbf{A}) = \sum_{\sigma \in S} \Pi_{k=1}^{S} A_{k\sigma_k}$, with $\sigma$ the vector of permutations of $(1,2,\ldots .S)$, denotes the permanent of the matrix $\mathbf{A}$, which is defined analogously to the determinant but does not have the alternating sign in its definition and therefore is much harder to calculate because, generally, $\mathrm{per}(\mathbf{AB}) \neq \mathrm{per}(\mathbf{A})\mathrm{per}(\mathbf{B})$. Permanent calculation is one of the prime examples of #$P$-hard problems in the field of computational complexity [26]. For an $n \times n$ matrix, the scaling of the numerical effort for its calculation via Ryser's formula [27] is of $\mathcal{O}(n^2 2^n)$, or $\mathcal{O}(n2^n)$ using Gray code [28], as compared to $\mathcal{O}(n^{2.373})$ for the determinant. We note that all these scalings are much better than that of Laplace expansion, which is $\mathcal{O}(nn!)$, but they are still exponentially expensive. The world record for permanent calculation has just been pushed to matrices of sizes as small as 54x54 [29].

To make an analogy with unitary time evolution, we write the rotation operator appearing in (1) as

$$\hat{R}\left(\mathbf{U^T}\right) = \exp\left(-\mathrm{i}\hat{H}\right), \tag{4}$$

with the help of the beam splitting Hamiltonian

$$\hat{H} = \hat{\mathbf{a}}^{\dagger \mathrm{T}} \boldsymbol{\Phi} \hat{\mathbf{a}}, \qquad \boldsymbol{\Phi} = \mathrm{i} \ln \mathbf{U}^T, \tag{5}$$

where $\hat{\mathbf{a}}$ denotes the column vector of annihilation operators on the $M$ modes and $\boldsymbol{\Phi}$, in general, is a full $M \times M$ Hermitian matrix with entries $\Phi_{ij}$ [30]. The action of the rotation operator on elements of the vector of creation operators is then given by

$$\hat{R}\left(\mathbf{U^T}\right) \hat{a}_i^\dagger \hat{R}\left(\mathbf{U^T}\right)^\dagger = \sum_j U_{ij} \hat{a}_j^\dagger, \tag{6}$$

which can be proven using the Baker-Haussdorff (or Hadamard) lemma [31] and which will be employed below. To generate the numerical results presented in Sec.4, we used the Matlab code for the creation of Haar random unitary (HRU) matrices provided by Cubitt [32].

## 3 Application of the HRU in the GCS Basis

Glauber (or standard) coherent state basis functions, which consist of a superposition of number states, are well suited for the dynamics of continuous variable systems that are close to harmonic [33], but are not ideal for the treatment of the quantum dynamics governed by lattice Hamiltonians (like in boson sampling) for systems with a fixed number $S$ of particles distributed among $M$ modes (sites). It has been realized recently, however, that generalized coherent states (GCS) [34], most favorably defined via

$$|S, \vec{\xi}\rangle = \frac{1}{\sqrt{S!}} \left( \sum_{i=1}^{M} \xi_i \hat{a}_i^\dagger \right)^S |0, 0, \ldots\rangle, \tag{7}$$

where $\sum_{i=1}^{M} |\xi_i|^2 = 1$ and with the many-particle vacuum state $|0, 0, \ldots\rangle$ and bosonic creation operators $\hat{a}_i^\dagger$ acting at the $i$-th site, are well-suited as basis function for the description of the particle conserving dynamics of systems like the Bose-Hubbard Hamiltonian [35–37]. A variational mean-field approach based on a single GCS has been proposed in [35] and a fully variational approach using a multi-configuration GCS-Ansatz has been formulated in [21].

### 3.1 Exact representation of the initial state: Kan summation formula

In the present manuscript, we take a direct approach to the unitary evolution in the boson sampling problem without invoking the variational principle. We do so by using the *multiconfigurational expansion* of the wavefunction in terms of the GCS defined above, according to

$$|\Psi(t)\rangle = \sum_{k=1}^{N} A_k(t)|S, \vec{\xi}_k(t)\rangle, \tag{8}$$

where $\vec{\xi}_k = (\xi_{k1}, \xi_{k2}, \dots, \xi_{kM})$ denotes a time-dependent vector of complex-valued GCS parameters and with multiplicity index $k$ ranging from 1 to $N$. In this general form of the wavefunction, both the expansion coefficients $\{A_k\}$ as well as the GCS parameters are in principle time-dependent and complex valued. We can, however, refrain from the explicit time-dependence of the coefficients, if the exact expansion of the initial state in terms of GCS is used, as will be shown below. Applying the HRU of boson sampling from Eq. (4) amounts to evolution over the complete unit time interval ($t = 1$), but below, we will also consider finite time evolution, which is characterized by an exponent of the unitary operator $\exp(-i\hat{H}t)$, where $t \in [0, 1]$. This will allow us to study the build-up of entanglement, similar to the case of dynamics after a quench [38].

For the applicability of the proposed approach, it is decisive that the initial Fock state can be represented in terms of GCS. This puzzle can be solved exactly analytically by the use of Kan's formula for monomials (a single product of powers) [23]

$$x_1^{s_1} x_2^{s_2} \cdots x_n^{s_n} = \frac{1}{S!} \sum_{\nu_1=0}^{s_1} \cdots \sum_{\nu_n=0}^{s_n} (-1)^{\sum_{i=1}^{n} \nu_i} \cdot \binom{s_1}{\nu_1} \cdots \binom{s_n}{\nu_n} \left( \sum_{i=1}^{n} h_i x_i \right)^S, \tag{9}$$

where $S = s_1 + s_2 + \dots s_n$, with integers $s_i \geq 0$ and $h_i = s_i/2 - \nu_i$. We observe that the term $\left( \sum_{i=1}^{n} h_i x_i \right)^S$ on the right hand side has a structure appearing also in the definition of the SU($M$) coherent states given in Eq. (7). By replacing the formal variables $x_i$ by creation operators $a_i^\dagger$, we are thus able to build the (exact) relationship between Fock states and SU($M$) coherent states according to

$$
\begin{aligned}
|s_1, s_2, \cdots, s_M\rangle &= \frac{1}{\sqrt{s_1! s_2! \cdots s_M!}} (a_1^\dagger)^{s_1} (a_2^\dagger)^{s_2} \cdots (a_M^\dagger)^{s_M} |00\cdots 0\rangle \\
&= \frac{1}{\sqrt{s_1! s_2! \cdots s_M!}\sqrt{S!}} \sum_{\nu_1=0}^{s_1} \cdots \sum_{\nu_M=0}^{s_M} (-1)^{\sum_{i=1}^{M} \nu_i} \\
&\qquad \binom{s_1}{\nu_1} \cdots \binom{s_M}{\nu_M} \left( \sum_{i=1}^{M} |h_i|^2 \right)^{S/2} \frac{1}{\sqrt{S!}} \left( \sum_{i=1}^{M} \xi_i a_i^\dagger \right)^S |00\cdots 0\rangle \\
&= \frac{1}{\sqrt{s_1! s_2! \cdots s_M!}} \sum_{k=1}^{(s_1+1)(s_2+1)\cdots(s_M+1)} A_k |S, \vec{\xi}_k\rangle.
\end{aligned}
\tag{10}
$$

The initial coefficients and parameters of the SU($M$) coherent states thus are

$$A_k = (-1)^{\sum_{i=1}^{M} \nu_{ki}} \binom{s_1}{\nu_{k1}} \cdots \binom{s_M}{\nu_{kM}} \frac{\left( \sum_{i=1}^{M} |h_{ki}|^2 \right)^{S/2}}{\sqrt{S!}}, \tag{11}$$

$$\vec{\xi}_k = \frac{1}{\sqrt{\sum_{i=1}^{M} |h_{ki}|^2}} (h_{k1}, h_{k2}, \cdots, h_{kM}), \tag{12}$$

where $\{h_{ki}\} = \{\frac{s_i}{2} - v_{ki}\}$ and the set of $\{v_{k1}, v_{k2}, \cdots, v_{kM}\}$ represents the $k$th possible combination of $v_1 = \{0, 1, \cdots, s_1\}$, $v_2 = \{0, 1, \cdots, s_2\}$, $\cdots$, $v_M = \{0, 1, \cdots, s_M\}$. The factor $\left(\sum_{i=1}^{M} |h_{ki}|^2\right)^{S/2}$ has been introduced in the definition of the coefficients in order to normalize the GCS parameter vectors.

For the special case $s_1 = s_2 = \cdots = s_S = 1$, we have ($S \leq M, N = 2^{S-1}$)

$$|11\cdots100\cdots0\rangle = \sum_{k=1}^{N} A_k |S, \vec{\xi}_k\rangle, \tag{13}$$

where $A_k/2 = (-1)^{\sum_{i=1}^{S} v_{ki}} \binom{1}{v_{k1}} \cdots \binom{1}{v_{kS}} \dfrac{\left(\sum_{i=1}^{S} |h_{ki}|^2\right)^{S/2}}{\sqrt{S!}}$, $\{h_{ki}\} = \{\frac{1}{2} - v_{ki}\}$, $v_1 = \{0\}$, $v_2 = \{0,1\}, \cdots, v_S = \{0,1\}$ and $\xi_{k,S+1} = \cdots = \xi_{k,M} = 0$. Here, we have used that there is a redundancy in Kan's original formula (9), already noticed by Kan himself. This results in the fact that we can reduce the multiplicity summation from $N = 2^S$ to $N = 2^{S-1}$ terms by fixing the first index $v_1$ to be zero. In passing, we note that the Kan formula underlying the present procedure thus involves an exponential scaling in the particle number of the required number of GCS basis functions. The numerical overhead in terms of mode number scales polynomially for a given number of particles, and thus we can handle a large number of modes efficiently. Overall, this is in clear contrast to the typically much more demanding factorial scaling, according to $(M + S - 1)!/[S!(M - 1)!]$, of the number of basis functions that would be required in a Fock space calculation.

## 3.2 Unitary evolution

With the boson sampling setup in mind, we now assume that the input state is the Fock state $|11\cdots100\cdots0\rangle$, which we just discussed, and that the linear optical circuit is described by the rotation operator $\hat{R}$ from Eq. (4). Using Eqs. (6,7) and (10), the output state (at $t = 1$) can then be written as

$$\begin{aligned}
|\Psi\rangle_{\text{out}} &= \hat{R}|11\cdots100\cdots0\rangle \\
&= \sum_{k=1}^{N} \frac{A_k}{\sqrt{S!}} \hat{R}\left(\sum_{i=1}^{M} \xi_{ki} a_i^\dagger\right)^S \hat{R}^{-1}\hat{R}|00\cdots0\rangle \\
&= \sum_{k=1}^{N} \frac{A_k}{\sqrt{S!}} \left(\sum_{i=1}^{M} \xi_{ki} \sum_{j=1}^{M} U_{ij} a_j^\dagger\right)^S |00\cdots0\rangle \\
&= \sum_{k=1}^{N} A_k |S, \sum_{i=1}^{M} \xi_{ki} U_{i1}, \sum_{i=1}^{M} \xi_{ki} U_{i2}, \cdots, \sum_{i=1}^{M} \xi_{ki} U_{iM}\rangle \\
&= \sum_{k=1}^{N} A_k |S, \vec{\xi}_k\rangle_{\text{out}}.
\end{aligned} \tag{14}$$

In contrast to an expansion in terms of Fock states, the GCS expansion coefficients (amplitudes) stay constant at all times and the GCS parameters $\{(\xi_{ki})_{\text{out}}\}$ characterizing the output state can be obtained by the matrix product

$$\begin{pmatrix} \vec{\xi}_1 \\ \vec{\xi}_2 \\ \vdots \\ \vec{\xi}_N \end{pmatrix}_{\text{out}} = \begin{pmatrix} \vec{\xi}_1 \\ \vec{\xi}_2 \\ \vdots \\ \vec{\xi}_N \end{pmatrix}_{\text{in}} \begin{pmatrix} U_{11} & U_{12} & \cdots & U_{1M} \\ U_{21} & U_{22} & \cdots & U_{2M} \\ \vdots & \vdots & \vdots & \vdots \\ U_{M1} & U_{M2} & \cdots & U_{MM} \end{pmatrix}. \tag{15}$$

In Appendix A, we show that the above output state allows for a rederivation of Glynn's formula for the permanent [22], given by

$$\text{per}(\mathbf{U}) = \langle 11 \cdots 1 | \Psi \rangle_{\text{out}}$$

$$= \frac{1}{2^{M-1}} \sum_{k=1}^{2^{M-1}} \left[ \left( \prod_{i=1}^{M} x_{ki} \right) \prod_{m=1}^{M} \vec{x}_k \cdot \vec{U}_m \right], \tag{16}$$

where $M = S$ and $\vec{U}_m$ is the $m$-th column vector of the matrix $\mathbf{U}$. The vector $\vec{x}_k$ is defined in the appendix.

## 4 The entanglement entropy

In the following, we investigate the creation of entanglement by application of the unitary operation of boson sampling. The initial Fock state of the unpartitioned system is not entangled because it can be written as a single product of single particle states. A single SU($M$) coherent state, in general, however, does not have this property, as can be inferred from its definition (7). Due to the important property of a single GCS being the ground state of the free-boson model, its entanglement properties have been studied in great depth by Dell'Anna using the reduced density matrix [14].

### 4.1 Bipartitioning and Rényi entropies

In contrast to the single GCS case, we now consider the multi-configuration case, with the state $|\Psi\rangle$ being the superposition of multiple SU($M$) coherent states, according to Eq. (8). Partitioning the system into a left and a right part, the (final) density operator of the full system is

$$\hat{\rho} = \sum_{k,j=1}^{N} A_k A_j^* |S, \vec{\xi}_k\rangle \langle S, \vec{\xi}_j|$$

$$= \frac{1}{S!} \sum_{n',n=0}^{S} \sum_{k,j=1}^{N} A_k A_j^* \binom{S}{n} \binom{S}{n'} \sqrt{n!(S-n)!n'!(S-n')!}$$

$$|S-n, \vec{\xi}_{k\tilde{L}}\rangle \langle S-n', \vec{\xi}_{j\tilde{L}}| \otimes |n, \vec{\xi}_{k\tilde{R}}\rangle \langle n', \vec{\xi}_{j\tilde{R}}|, \tag{17}$$

where the sum in the definition of the GCS, Eq. (7), was split in two parts ($L$ and $R$), a binomial expansion was used, and where the two non-normalized SU($M$) states are defined by

$$|S-n, \vec{\xi}_{\tilde{L}}\rangle = \frac{1}{\sqrt{(S-n)!}} \left( \sum_{i=1}^{M_L} \xi_i a_i^\dagger \right)^{S-n} |00 \cdots 0\rangle, \tag{18}$$

$$|n, \vec{\xi}_{\tilde{R}}\rangle = \frac{1}{\sqrt{n!}} \left( \sum_{i=M_L+1}^{M} \xi_i a_i^\dagger \right)^{n} |00 \cdots 0\rangle. \tag{19}$$

The fact that these GCS are not normalized is indicated by the tilde over the symbols $L$ as well as $R$.

The reduced density matrix of the left subsystem can then be derived by tracing over the right one. Using the fact that GCS with different particle numbers are orthogonal to each other,

this yields

$$
\begin{aligned}
\hat{\rho}_L &= \mathrm{Tr}_R(\hat{\rho}) \\
&= \sum_{n=0}^{S} \binom{S}{n} \sum_{k,j=1}^{N} A_k A_j^* \langle n, \vec{\xi}_{j\tilde{R}} | n, \vec{\xi}_{k\tilde{R}} \rangle | S-n, \vec{\xi}_{k\tilde{L}} \rangle \langle S-n, \vec{\xi}_{j\tilde{L}} |.
\end{aligned}
\tag{20}
$$

Going into Fock space, the density matrix corresponding to the above density operator, analogous to Eq. (5) in [14], can thus be written in block diagonal form

$$
\boldsymbol{\rho}_L = \mathrm{diag}\left(\boldsymbol{\rho}_L^{(0)}, \boldsymbol{\rho}_L^{(1)}, \cdots, \boldsymbol{\rho}_L^{(S)}\right),
\tag{21}
$$

where the uncoupled blocks $\boldsymbol{\rho}_L^{(n)}$ describe a distribution of $n$ particles on the right side and $S-n$ particles on the the left side of the cut. The elements of the blocks are given by

$$
\boldsymbol{\rho}_L^{(n)} = \binom{S}{n} \sum_{k,j=1}^{N} C_{jk}^{(n)} \vec{W}_k^{(n)} \vec{W}_j^{(n)\dagger},
\tag{22}
$$

with the coefficient $C_{jk}^{(n)} = A_k A_j^* \langle n, \vec{\xi}_{j\tilde{R}} | n, \vec{\xi}_{k\tilde{R}} \rangle$ and where the vector $\vec{W}_k^{(n)}$ is filled by the entries $\langle n_1 n_2 \cdots n_{M_L} | S-n, \vec{\xi}_{k\tilde{L}} \rangle$ with $\sum_{i=1}^{M_L} n_i = S-n$. The product of the column vector with the row vector is a matrix but in the reverse order (row times column) it is the scalar

$$
\vec{W}_j^{(n)\dagger} \cdot \vec{W}_k^{(n)} = \langle S-n, \vec{\xi}_{j\tilde{L}} | S-n, \vec{\xi}_{k\tilde{L}} \rangle = \left( \sum_{i=1}^{M_L} \xi_{ji}^* \xi_{ki} \right)^{S-n},
\tag{23}
$$

where, for the overlap between GCS, a formula from the appendix of [21] has been used. Due to ease of computation, as a measure of the entanglement of the left subsystem after the application of the unitary matrix of boson sampling, the linear entropy $S_L = 1 - \mathrm{Tr}(\boldsymbol{\rho}_L^2)$ [25] as well as the second order Rényi entropy (see below) are frequently used. For computational purposes, for the purity, it is favorable to use the expression

$$
\begin{aligned}
\mathrm{Tr}(\boldsymbol{\rho}_L^2) &= \sum_{n=0}^{S} \binom{S}{n}^2 \mathrm{Tr}\Big( \sum_{k,j=1}^{N} C_{jk}^{(n)} \vec{W}_k^{(n)} \vec{W}_j^{(n)\dagger} \sum_{k',j'=1}^{N} C_{j'k'}^{(n)} \vec{W}_{k'}^{(n)} \vec{W}_{j'}^{(n)\dagger} \Big) \\
&= \sum_{n=0}^{S} \binom{S}{n}^2 \sum_{k,j,k',j'=1}^{N} C_{jk}^{(n)} C_{j'k'}^{(n)} \mathrm{Tr}\Big( \vec{W}_k^{(n)} \vec{W}_j^{(n)\dagger} \vec{W}_{k'}^{(n)} \vec{W}_{j'}^{(n)\dagger} \Big) \\
&= \sum_{n=0}^{S} \binom{S}{n}^2 \sum_{k,j,k',j'=1}^{N} C_{jk}^{(n)} C_{j'k'}^{(n)} \left[ \vec{W}_j^{(n)\dagger} \vec{W}_{k'}^{(n)} \right] \left[ \vec{W}_{j'}^{(n)\dagger} \vec{W}_k^{(n)} \right] \\
&= \sum_{n=0}^{S} \binom{S}{n}^2 \sum_{k,j,k',j'=1}^{N} C_{jk}^{(n)} C_{j'k'}^{(n)} \left[ \left( \sum_{i=1}^{M_L} \xi_{ji}^* \xi_{k'i} \right) \left( \sum_{i=1}^{M_L} \xi_{j'i}^* \xi_{ki} \right) \right]^{S-n},
\end{aligned}
\tag{24}
$$

where we have used that the trace of the product of two operators amounts to a scalar product (step from line two to line three). This result can be specified to the case of the boson sampling problem by expressing the GCS parameters after the application of the HRU in terms of the Hermitian matrices

$$
\Lambda_L = (\vec{\mathcal{U}}_1, \vec{\mathcal{U}}_2, \cdots, \vec{\mathcal{U}}_{M_L}) \cdot (\vec{\mathcal{U}}_1, \vec{\mathcal{U}}_2, \cdots, \vec{\mathcal{U}}_{M_L})^\dagger,
\tag{25}
$$

$$
\Lambda_R = (\vec{\mathcal{U}}_{M_L+1}, \vec{\mathcal{U}}_{M_L+2}, \cdots, \vec{\mathcal{U}}_M) \cdot (\vec{\mathcal{U}}_{M_L+1}, \vec{\mathcal{U}}_{M_L+2}, \cdots, \vec{\mathcal{U}}_M)^\dagger,
\tag{26}
$$

derived in Appendix B and where $\vec{\mathcal{U}}_i$ is the truncated vector $\vec{U}_i$ with only the first $S$ entries. There also the purity is reexpressed as

$$\text{Tr}(\boldsymbol{\rho}_L^2) = \left(\frac{1}{2}\right)^{4(S-1)} \sum_{n=0}^{S} \frac{1}{[(S-n)!n!]^2}$$
$$\sum_{k,j,k',j'=1}^{2^{S-1}} \prod_{i=1}^{S} \left(x_{ki}x_{ji}x_{k'i}x_{j'i}\right) \left(\vec{x}_{k'}\Lambda_L\vec{x}_j^T \vec{x}_k\Lambda_L\vec{x}_{j'}^T\right)^{S-n} \left(\vec{x}_k\Lambda_R\vec{x}_j^T \vec{x}_{k'}\Lambda_R\vec{x}_{j'}^T\right)^n . \quad (27)$$

This formula allows us to write the purity in terms of particle as well as mode number and the entries of the unitary matrix $U$ only, not making reference to GCS any longer.

The previous form of the purity, given in Eq. (24), with the output GCS parameters $\{\vec{\xi}_k\}$ from Eq. (15), however, is better suited for numerical purposes, as it can be calculated easily in matrix language. Also, there is no need to calculate the eigenvalues of the reduced density matrix for typically huge Fock space dimensions, which would be necessary to calculate the von Neumann entropy

$$S_{\text{vN}} = -\text{Tr}\left(\boldsymbol{\rho}_L \ln \boldsymbol{\rho}_L\right) = -\sum_{n=0}^{S} \text{Tr}\left(\boldsymbol{\rho}_L^{(n)} \ln \boldsymbol{\rho}_L^{(n)}\right). \quad (28)$$

In the focus of the results section below thus are the so-called Rényi entropies

$$S_\alpha = \frac{1}{1-\alpha} \ln \text{Tr}(\boldsymbol{\rho}_L^\alpha). \quad (29)$$

For $\alpha = 2$ the purity just discussed is needed for its calculation. In the general case, higher powers of the density matrix have to be considered, however. This can be done along lines, similar to those of Eq. (24). The quadruple sum over the multiplicity indices would, e.g., turn into a sextupel sum for $\alpha = 3$. We stress that the number of terms in every sum can be reduced to $2^{S-1}$ by using the non-redundant Kan formula mentioned above. A relation between the Rényi entropy and the von Neuman entropy that we will refer to below is $S_{\text{vN}} = \lim_{\alpha \to 1} S_\alpha$, and it is well-known that $S_1 \geq S_2 \geq S_3 \dots$ [39]; see also [40] for a recent discussion of Rényi entropy inequalities in the context of Gaussian boson sampling. For the one-dimensional XY spin chain some useful exact results for Rényi entropies as a function of $\alpha$ are available [41]. Rényi entropies for noninteger values $0 < \alpha < 1$ are discussed in [42]. An experimental approach to measuring Rényi entropies employs the preparation of two copies of the same system and measuring the expectation value of the so-called swap operator. It has originally been devised to investigate quench dynamics in Bose-Hubbard type lattice Hamiltonians by Daley et al. [43].

## 4.2 Numerical results for the case $t = 1$

We first focus on the case $t = 1$ of time evolution with the full unitary matrix. All the entropies are calculated by averaging over different realizations of the HRU matrix, analogous to the analytic work by Page [44]. Individual realizations (not shown) do not differ much from the plots to be shown below, however. Firstly, in Figure 1, the Rényi entropy for index $\alpha = 2$ as a function of subsystem size, corresponding to the locus of the bipartition, is depicted for different photon numbers, ranging between 8 and 12 (single realization calculations for 14 photons (not shown) are also possible within a few hours on a standard workstation). The total number of modes in all cases is 200, and the displayed Page curves are all symmetric around bipartition mode number 100, in contrast to the asymmetric curves in [11]. For a

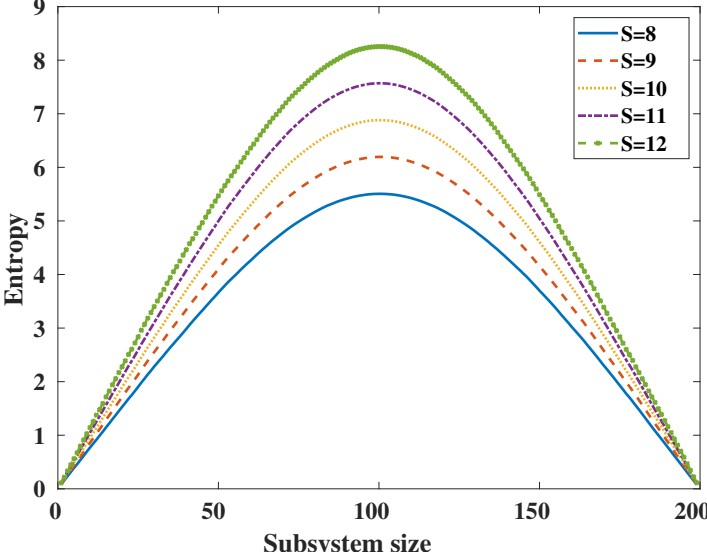

Figure 1: Rényi entropy (averaged over 100 realizations of the HRU) with index $\alpha = 2$ as a function of the size of the left system for different photon numbers, ranging from $S = 8$ to $S = 12$, in the case of a fixed number of modes ($M = 200$).

subsystem size smaller than $S$, the entropy follows a volume law (see also the discussion in the following subsection), and the maximum entropy [1] is displayed for splitting the system into two equal halves. The scaling of this maximum entropy with particle number is linear. We stress that the Rényi entropy is a lower bound for the von Neumann entropy, and our exact results can be taken as a benchmark for other, purely numerical calculations. Furthermore, we stress that we can observe asymmetric curves for the entropy if we consider exponents $t < 1$, as will be discussed in the following subsection.

Secondly, the Rényi entropy for different index $\alpha$ as a function of subsystem size in a system with $S = 10$ and $M = 500$ is depicted in Fig. 2. Again, for subsystem size smaller than $S$, a volume law is found with the slope depending on the Rényi index. Changing the number of photons does not qualitatively alter the results. They show that by increasing $\alpha$, the entropy is monotonically decreasing (in complete agreement with classical results from symbolic dynamics [39]), and the functional form turns from concave to (almost) convex (the second order derivative (assuming the abscissa to be a continuous variable) of the blue curve in Fig. 2 is negative, whereas the second derivative of the green curve is almost everywhere positive).

Interestingly, the maximum of the entropy (at $M = 250$) for $S = 10$ is only slightly dependent on the Rényi index but we stress that the deviations of the maximum are not finite size effects, because, for different index the maximum entropy will increase linearly with the particle number (with an decreasing slope for increasing $\alpha$) , when the mode number is very large (not shown). The fact that we can calculate Rényi entropies for many different values of $\alpha$ would allow us also to study entanglement spectra [45].

Finally, we studied the maximum Rényi entropy (at equal partition) as a function of the total mode number. All the curves shown are almost saturating with increasing $M$, so the maximum entropy tends to stay constant from a certain size. We checked the case of $M > 1000$ and still found a slight increase, though. This finding corroborates the assumption that in order to fully reach the so-called collision-free subspace limit, it might be necessary that $M \geq S^5 \log^2 S$ [24]. Thus, in essence, the finding of this last numerical result is that, un-

---

[1]Defined as the maximal value of the entanglement entropy as a function of subsystem size (after averaging).

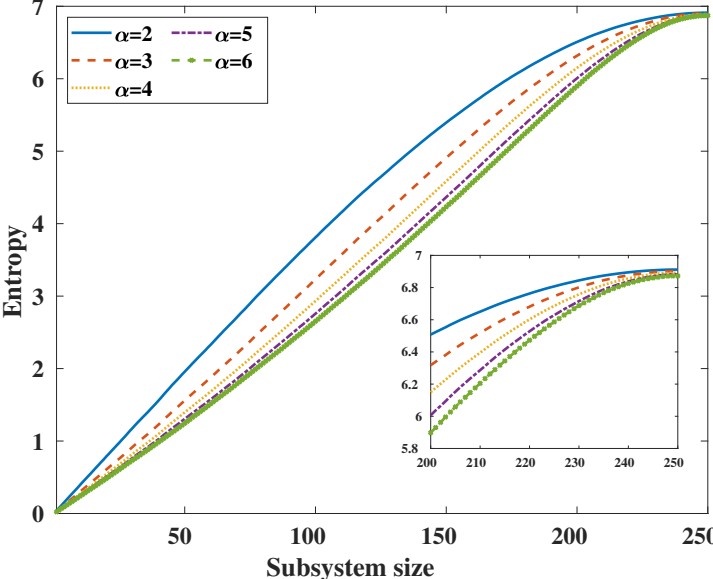

Figure 2: Rényi entropy (averaged over 100 realizations of the HRU) as a function of bipartition mode number for different $\alpha$ for fixed $S = 10$ and $M = 500$. Only the left side of the symmetric curve is depicted. The inset shows more clearly that the curves do not converge to the exact same maximum.

der collision-free subspace conditions, the maximum entropy that the unitary application can gain is achieved. We also stress that the initial quick increase of the maximum entropy, which we observe for small mode numbers and which is also seen in a similar setup [46] is broken for higher mode numbers. The scaling of the maximum entropy with respect to particle number can be extracted from Figure 1. This scaling is linear, corroborating the finding displayed in the last entry of Table I in [47], but in contrast to the observation of a logarithmic scaling in [13] for a nonlinear optical network. It is also worthwhile to note that although there is still an increase in entanglement for large mode numbers (at fixed particle number), the numerical effort to produce the results is not growing exponentially with $M$ since the number of basis functions is not changing.

## 4.3 Build-up of entanglement

In this last section, we mimic the build-up of entanglement by choosing the power of the unitary matrix (which in the previous section was given by $t = 1$) to be somewhere inside the interval $t \in [0, 1]$. The parameter $t$ then plays a similar role as does time in a study of the dynamics of a many-body system after a quench (sudden change of the Hamiltonian) [38]. In panels (a) and (b) of Fig. 4, where we took different roots of the HRU, it is revealed that for small values of $t$, the diagonal elements of the resulting matrix (a single matrix is displayed, we did not take an average) are still dominant, whereas for $t$ closer to unity, the structure along the diagonal gets washed out. For the case $t = 0.8$, displayed in panel (b), it is barely visible any longer, as the matrix elements tend to be fully random. The result of applying the evolution matrix to the initial state, according to Eq. (15), is represented in panels (c) and (d) of the same figure. Increasing $t$ leads to a transfer of population to mode numbers that are further and further away from the initially occupied ones. Thus at $t = 0.8$ a more even distribution of the population over all modes is observed.

The corresponding results for the entanglement shown in Fig. 5 are similar to the ones in [48] as they show a linear increase with particle number of the maximum entropy in the

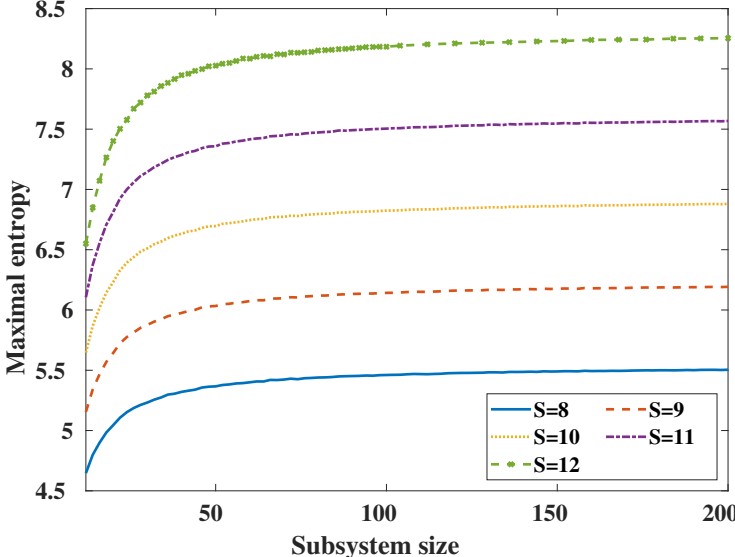

Figure 3: Maximum Rényi entropy ($\alpha = 2$) averaged over 500 realizations of the HRU as a function of mode number for different particle numbers $S$.

asymptotic regime. However, our results do not show an almost linear increase with time [38, 49] before the asymptotic regime is reached. This may be because a local coupling is employed to generate the results in [38]. In contrast, in our case, for all $t < 1$, the coupling is still fully nonlocal (taking the logarithm of the unitary matrices). In our results, a quadratic increase is followed by a linear one, and the maximum entropy is reached already at $t = 0.45$ regardless of the particle number. Thus, although the population has not been distributed evenly across the system (see Fig. 4), the entropy has already reached its maximum.

Finally, in Fig. 6 we show the full entropy curve as a function of subsystem size for different values of $t$ for $S = 10$. The graphs display an asymmetric shape with a cusp-like structure at bipartition mode number equal to the particle number $S = 10$, due to the choice of the asymmetric initial state. The cusp gets smoothened out and the curves become more and more symmetric for larger values of $t$. At $t = 1$ there is no "memory" of the unsymmetric initial state left. For bipartition mode numbers smaller than ten, we always find a linear increase of the entropy, corresponding to a volume law, as already mentioned in the discussion of Fig. 1. As also mentioned earlier, the maximum entropy is reached already at $t = 0.45$. It is, however, not located at symmetric bipartition if the exponent is not equal to unity. For $t \ll 1$, the correlation between the left (initially populated) and the right (initially unpopulated) part is not fully developed; thus, the entanglement will decrease for bipartition mode numbers above 10. For $t \to 0$, as expected, the entropy will shrink to an overall constant value of zero without a cusp.

## 5 Conclusions and Outlook

Using Kan's formula to express a Fock initial state in terms of generalized coherent states, we have derived an exact analytical formula for the output state of a linear optical network for standard boson sampling, given in terms of a finite sum over basis function multiplicity. The scaling of the numerical effort to evaluate the sum is exponential in particle number $S$ via $2^{S-1}$, but only polynomial in the number of modes $M$. In total, the computational complexity thus increases much less severely than the super-exponential scaling of the size of the Fock state

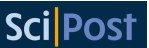

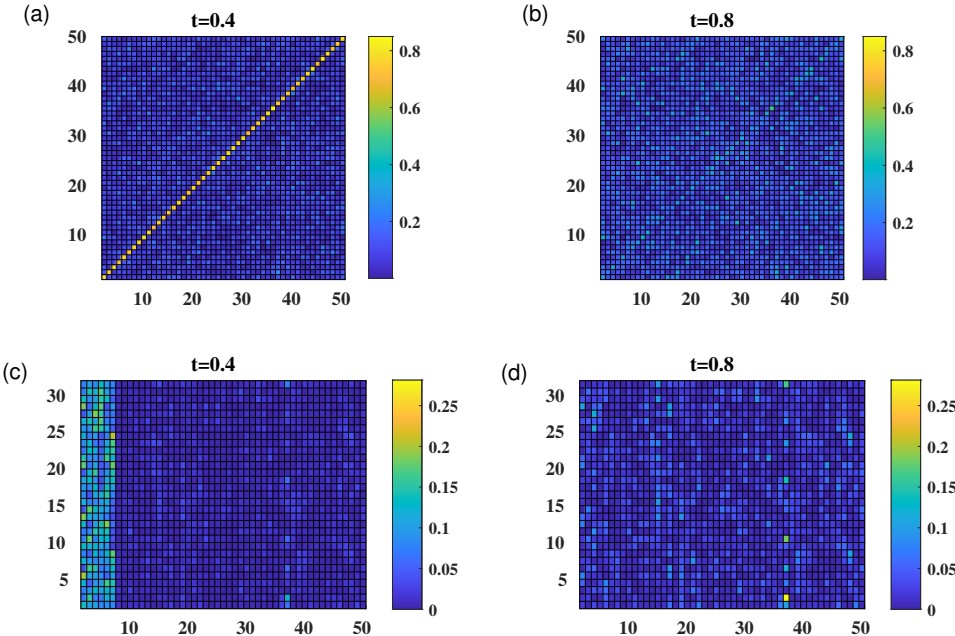

Figure 4: Panels (a) and (b): absolute values of the elements of different powers $t$ of the HRU matrix with $M = 50$; panels (c) and (d): absolute value squared of the GCS parameters $\xi_{ki}$ as a function of mode number ($x$-axis running from 1 to $M$) and multiplicity index ($y$-axis for $S = 6$ running to $N = 2^{6-1} = 32$).

Hilbert space, which is of dimension $(M + S - 1)!/[S!(M - 1)!]$. The tractable dependence on the mode number of the numerical effort of sum evaluation has allowed us to investigate the so-called collision-free subspace case, for which it is believed that $M \approx S^2$ is sufficient [24].

At the initial investigation stage, the output state wavefunction was derived in terms of multiple GCS. Along the way, we have also rederived the formula of Glynn for the permanent of a square matrix. Using the binomial theorem, the reduced density operator was calculated from the wavefunction. Employing this important intermediate result, the purity and traces of powers higher than two of the reduced density matrix were calculated exactly analytically and are given in terms of multiple sums that could probably be simplified further. The results involve overlaps of GCS and do not need the evaluation of eigenvalues of matrices for huge Hilbert space dimensions. This allowed us to study the creation of subsystem entanglement by applying the unitary matrix of boson sampling. In case of applying the full unitary matrix, i.e., for $t = 1$, we have corroborated numerically that the Rényi entropy is a decreasing function of the Rényi index and that the maximum Rényi entropy is realized at equipartition, regardless of the asymmetric population of the initial state. In addition, we found that the maximum entropy is only slightly dependent on the index $\alpha$. Furthermore, because of the polynomial dependence on $M$ of our numerical complexity, we could corroborate that under collision-free subspace conditions, the maximum Rényi entropy saturates as a function of mode number. Finally, a cusp in the (generally asymmetric) entropy curve at partition mode number equal to particle number was found by investigating the build-up of entanglement. As an important insight of the last subsection, it turned out that although the population has not yet fully equilibrated, the entropy already has reached its maximum value at $t \approx 0.45$.

By using an exact analytical expression for the time-evolved wavefunction, without invoking any approximation (like reduced bond dimension in MPS), we have thus been able to

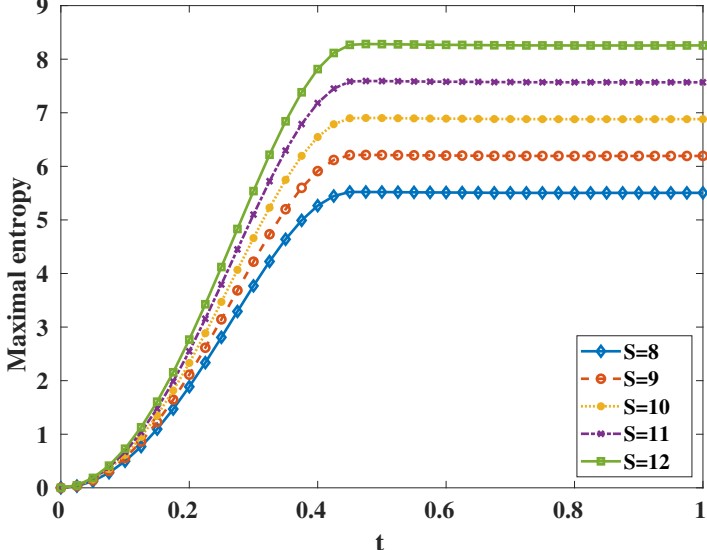

Figure 5: Maximum Rényi entropy ($\alpha = 2$) as a function of $t$ (time) averaged over 100 realizations of the unitary for different particle numbers $S$ and fixed mode number $M = 200$.

study strongly entangled system dynamics on a classical computer, which is considered to be challenging [19]. This was possible because single elements of the GCS basis we used are already highly entangled, which is not the case with single Fock states or MPS calculations. Furthermore, we stress that in our simulation, we have the full state vector at our disposal. This has allowed us to study bipartite entanglement.

In future work, the time-scale at which the entropy reaches its maximum shall be investigated in more detail, e.g., with respect to its dependence on system size and/or mode number. Furthermore, it would be worthwhile to also look into calculating other dynamical quantities, like, e. g., higher-order correlation functions [50], using generalized coherent states. Furthermore, apart from optical circuits, other circuits, like fermionic ones or molecular vibronic spectroscopy [51], may be investigated using our current approach. In addition, it would be a promising task to further manipulate the analytical results for the purity to simplify and better understand them. These considerations may also enlighten the relationship between entanglement and computational complexity. In addition, the open question why entanglement is growing to its maximum value surprisingly fast deserves further attention. Finally, the extension of the present investigations to Gaussian boson sampling [40, 52, 53], might be worthwhile, with a special focus on the time-scale mentioned above.

# A Rederivation of Glynn's formula for the permanent

We verify the output wavefunction of Fock state boson sampling given in Eq. (14) by showing that it contains Glynn's formula for the permanent of a square matrix. To this end, we stress that, in order to calculate the permanent of the unitary matrix $\mathbf{U}$, we can assume the input state to be the special Fock state $|11\cdots1\rangle$. Then, from Eq. (12) the parameters of the SU($M$) coherent states are given by $\vec{\xi}_k = \frac{1}{\sqrt{M}}\vec{x}_k$, where $\vec{x}_k$ is a vector with $M$ entries chosen from the set $\{-1, +1\}$, except for $k = 1$, where the vector entries are fixed to be $+1$, and the

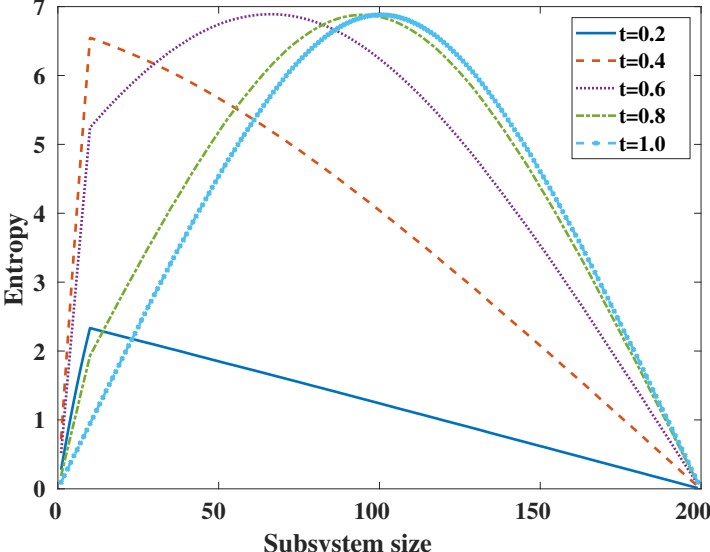

Figure 6: Maximum Rényi entropy ($\alpha = 2$) averaged over 100 realizations of the HRU as a function of mode number for different depths and fixed particle number $S = 10$.

corresponding amplitude from Eq. (11) is given by

$$A_k = \frac{1}{\sqrt{M!}} \left(\frac{M}{4}\right)^{\frac{M-1}{2}} \prod_{i=1}^{M} x_{ki} \,. \tag{30}$$

So the output state under the unitary transformation is (presently $S = M$)

$$|\Psi\rangle_{\text{out}} = \frac{1}{\sqrt{M!}} \left(\frac{M}{4}\right)^{\frac{M-1}{2}} \sum_{k=1}^{2^{M-1}} \left(\prod_{i=1}^{M} x_{ki}\right) \left| M, \frac{1}{\sqrt{M}}\vec{x}_k \cdot \vec{U}_1, \frac{1}{\sqrt{M}}\vec{x}_k \cdot \vec{U}_2, \cdots, \frac{1}{\sqrt{M}}\vec{x}_k \cdot \vec{U}_M \right\rangle, \tag{31}$$

where $\vec{U}_j$ denotes the $j$-th column of the matrix $\mathbf{U}$. The permanent of the unitary matrix can then be obtained by projecting the output state onto the Fock state $\langle 11 \cdots 1|$, yielding

$$\begin{aligned}
\text{per}(\mathbf{U}) &= \langle 11 \cdots 1 | \Psi \rangle_{\text{out}} \\
&= \frac{1}{2^{M-1}} \sum_{k=1}^{2^{M-1}} \left[ \left(\prod_{i=1}^{M} x_{ki}\right) \prod_{m=1}^{M} \vec{x}_k \cdot \vec{U}_m \right],
\end{aligned} \tag{32}$$

where we have used the multinomial theorem and the fact that only terms with unit powers of $a_i^\dagger$ do survive the projection. The resulting formula thus is Glynn's formula for the permanent of a square matrix [22,54].

Our analytical manipulations based on Kan's formula for the expansion of a Fock state in terms of multiple GCS thus lead to an alternative derivation of Glynn's formula, which is considered to be a computational alternative to Ryser's formula [27]. A generalized formula for the permanent has been found along similar lines [55,56]. Furthermore, it is worthwhile to note that different permanent identities have been proven in a quantum-inspired way in [57]. There the Glynn formula has, e.g., been proven using cat states.

# B  Derivation of the purity in terms of the unitary

For the boson sampling problem, if the initial state is the Fock state $|11\cdots10\cdots0\rangle$, where only the first $S$ modes are occupied by single photons, in close analogy to the rederivation of Glynn's formula in the previous appendix, Kan's formula implies that the values of the amplitudes in the GCS expansion in Eq. (8) are $A_k = \frac{1}{\sqrt{S!}}(\frac{S}{4})^{\frac{S-1}{2}}\prod_{i=1}^{S}x_{ki}$ where $\vec{x}_k$ is a vector with $S$ entries from the set $\{-1,+1\}$, apart from $k=1$, and the values of the GCS parameters are $\vec{\xi}_k = \frac{1}{\sqrt{S}}(x_{k1},x_{k2},\cdots,x_{kS},0,\cdots,0)$.

In analogy to the case $S=M$ from Eq. (31), the output state after the rotation with the unitary matrix $\mathbf{U}$ is

$$|\psi\rangle_{\text{out}} = \frac{1}{\sqrt{S!}}\left(\frac{S}{4}\right)^{\frac{S-1}{2}}\sum_{k=1}^{2^{S-1}}\left(\prod_{i=1}^{S}x_{ki}\right)\left|S,\frac{1}{\sqrt{S}}\vec{x}_k\vec{\mathcal{U}}_1,\frac{1}{\sqrt{S}}\vec{x}_k\vec{\mathcal{U}}_2,\cdots,\frac{1}{\sqrt{S}}\vec{x}_k\vec{\mathcal{U}}_M\right\rangle, \tag{33}$$

where $\vec{\mathcal{U}}_i$ is the truncated vector $\vec{U}_i$ with only the first $S$ entries. Thus, we get ($A$ coefficients are time-independent)

$$A_k A_j^* = \frac{1}{S!}\left(\frac{S}{4}\right)^{S-1}\prod_{i=1}^{S}(x_{ki}x_{ji}), \tag{34}$$

as well as

$$\begin{aligned}\langle S-n,\vec{\xi}_{j\tilde{L}}|S-n,\vec{\xi}_{k'\tilde{L}}\rangle_{\text{out}} &= \left(\vec{\xi}_{k'\tilde{L}}\vec{\xi}_{j\tilde{L}}^{\dagger}\right)^{S-n}\\ &= \left[\frac{1}{\sqrt{S}}\vec{x}_{k'}\cdot(\vec{\mathcal{U}}_1,\vec{\mathcal{U}}_2,\cdots,\vec{\mathcal{U}}_{M_L})\cdot\frac{1}{\sqrt{S}}(\vec{\mathcal{U}}_1,\vec{\mathcal{U}}_2,\cdots,\vec{\mathcal{U}}_{M_L})^{\dagger}\cdot\vec{x}_j^T\right]^{S-n}\\ &= \frac{1}{S^{S-n}}(\vec{x}_{k'}\Lambda_L\vec{x}_j^T)^{S-n}, \end{aligned}\tag{35}$$

and

$$\begin{aligned}\langle n,\vec{\xi}_{j\tilde{R}}|n,\vec{\xi}_{k\tilde{R}}\rangle_{\text{out}} &= \left(\vec{\xi}_{k\tilde{R}}\vec{\xi}_{j\tilde{R}}^{\dagger}\right)^{n}\\ &= \left[\frac{1}{\sqrt{S}}\vec{x}_k\cdot(\vec{\mathcal{U}}_{M_L+1},\vec{\mathcal{U}}_{M_L+2},\cdots,\vec{\mathcal{U}}_M)\cdot\frac{1}{\sqrt{S}}(\vec{\mathcal{U}}_{M_L+1},\vec{\mathcal{U}}_{M_L+2},\cdots,\vec{\mathcal{U}}_M)^{\dagger}\cdot\vec{x}_j^T\right]^{n}\\ &= \frac{1}{S^{n}}(\vec{x}_k\Lambda_R\vec{x}_j^T)^{n}, \end{aligned}\tag{36}$$

for the ingredients of Eq. (22). Here the Hermitian matrices $\Lambda_{L/R}$ are defined as

$$\Lambda_L = (\vec{\mathcal{U}}_1,\vec{\mathcal{U}}_2,\cdots,\vec{\mathcal{U}}_{M_L})\cdot(\vec{\mathcal{U}}_1,\vec{\mathcal{U}}_2,\cdots,\vec{\mathcal{U}}_{M_L})^{\dagger}, \tag{37}$$

$$\Lambda_R = (\vec{\mathcal{U}}_{M_L+1},\vec{\mathcal{U}}_{M_L+2},\cdots,\vec{\mathcal{U}}_M)\cdot(\vec{\mathcal{U}}_{M_L+1},\vec{\mathcal{U}}_{M_L+2},\cdots,\vec{\mathcal{U}}_M)^{\dagger}. \tag{38}$$

This leads to the final result

$$\begin{aligned}\text{Tr}(\boldsymbol{\rho}_L^2) = \left(\frac{1}{2}\right)^{4(S-1)}\sum_{n=0}^{S}\frac{1}{[(S-n)!n!]^2}\\ \sum_{k,j,k',j'=1}^{2^{S-1}}\prod_{i=1}^{S}(x_{ki}x_{ji}x_{k'i}x_{j'i})\left(\vec{x}_{k'}\Lambda_L\vec{x}_j^T\vec{x}_k\Lambda_L\vec{x}_{j'}^T\right)^{S-n}\left(\vec{x}_k\Lambda_R\vec{x}_j^T\vec{x}_{k'}\Lambda_R\vec{x}_{j'}^T\right)^{n}, \end{aligned}\tag{39}$$

for the purity.

# Acknowledgments

JH acknowledges fruitful discussions with Prof. V. Man'ko. FG would like to thank Prof. Jan Budich and Prof. Hong-Hao Tu for fruitful discussions and the International Max Planck Research School on Many Particle Systems in Structured Environments for its support.

**Funding information** This work was supported by Basic Science Research Program through the National Research Foundation of Korea (NRF) funded by the Ministry of Education, Science and Technology (NRF-2021M3E4A1038308, NRF-2021M3H3A1038085, NRF-2019M3E4A1079666, NRF-2022M3H3A106307411).

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
