# Peer review of "Entanglement in the full state vector of boson sampling"

_SciPost Physics, doi:SciPost Phys. 15, 007 (2023)_

## Round 1 · Referee Report · Anonymous (Referee 1) · 2023-3-28

Strengths

The paper presents a new application of a well known approach to improve the computation efficiency for a notoriously hard problem.

Weaknesses

No discussion of possible experimental realisation, non citations of boson sampling experiments (a lot with photons!).

Report

The paper reproposes the SU(M) approach to the Bose-Hubbard model, in particular useful for 2 sites for practical purposes. This approach is not new, see earlier papers by Korsch (cited here), Brandes (Dicke type of models), and Milburn (milestone paper 1997), but offers a new application to more efficiently compute the temporal evolution of such many-body quantum systems and its correlations. The analysis is exhaustive and the results, in particular on the entropy dynamics are interesting. My suggestion would be to add a short discussion about 1) experiments performed already on Bose sampling etc. and 2) experimental possibilities in implementing the research studied here in detail.

Requested changes

Please see my report.

  • validity: top
  • significance: high
  • originality: high
  • clarity: top
  • formatting: excellent
  • grammar: good

Author:  Yulong Qiao  on 2023-04-11  [id 3571]

(in reply to Report 2 on 2023-03-28)
Category:
remark

Dear Editor-in-charge,

herewith, on behalf of my coauthors, I would like to reply to the comments in Report 2 of the manuscript entitled

"Entanglement in the full state vector of boson sampling"

which we had submitted for publication as an article in SciPost Physics.

The reviewer had proposed to engage more on the following two points:

1) experiments performed already on boson sampling etc.

2) experimental possibilities in implementing the research studied here in detail.

We thank the Referee for the positive assessment of our submission. In a revised version of the manuscript, we would add a short paragraph addressing the issues mentioned above. With respect to the first point, we would reference to the article by D.J. Brod et al. in Advanced Photonics {\bf 1}, 034001 (2019) and references 63-67 and 69, 76 therein.

As to point two raised above, there is recent work on Bose-Hubbard type systems, realized in the so-called quantum gas microscope that addresses the measurement of Renyi entropies [A. J. Daley et al., ``Measuring entanglement growth in quench dynamics of bosons in an optical lattice'', Phys. Rev. Lett. 109, 020505 (2012)]. This scheme is designed for lattice-type bosonic system and thus might also be applicable to the boson sampling setup, where the role of the sites is played by the different modes. A brief discussion of the formalism that relies on preparing two copies of the same system and measuring the expectation value of the so-called swap operator will be added on resubmission.

Sincerely yours, Yulong Qiao

Anonymous on 2023-04-21  [id 3605]

(in reply to Yulong Qiao on 2023-04-11 [id 3571])

My comments have all been addressed successfully. I suggest publication now.

---

## Round 1 · Referee Report · Anonymous (Referee 2) · 2023-4-3

Report

In the manuscript titled "Entanglement in the full state vector of boson sampling", the authors give analytic expressions for the entanglement of a linear-optical state that is the output of an M × M linear-optical unitary applied to a Fock state with S bosons. This expression can be evaluated on a computer with runtime scaling polynomially in the number of modes M, although it scales exponentially in the number of bosons as 2ˢ⁻¹. The authors first outline the generalised coherent states (GCS) basis and explain how Fock states can be expressed in terms of this basis through the Kan formula. Then, the authors show how this is updated when a Haar-random unitary is applied. They then study entanglement properties of the output state, particularly the Rényi entropy with index α.

The first result is for the so-called Page curve of the Rényi entropy, which is in line with the expectation. The authors also study aspects of the maximum Rényi entropy as a function of the number of modes M, the number of particles S, and the Rényi index α. Finally, they study these quantities when one applies the t'th power of a Haar-random unitary (0 ≤ t ≤ 1). This simulates continuous-time evolution of the bosonic system in a system with no locality constraints. The authors give interesting results on the growth of entanglement in time and the growth profile in space. They find that often, the maximum value of the entanglement is reached by a time t ~0.45. However, the entanglement profile as well as the distribution of bosons both significantly differ from their values at t=1. I think there is a nice interpretation here that the authors alluded to but have not expanded on. I believe this is because at the timescale of t~0.45, most of the initial Fock state bosons have "split" into a superposition over two modes, leading to the entanglement being near maximal. However, it appears that one needs to evolve for some more time until this superposition is further spread out in space. One more comment I have for the authors is to study this timescale in more detail: is it a system-size independent time scale? Does it depend strongly on the number of modes? Is there a phase transition associated with this behaviour? And is the behaviour different if one considers not a Haar-random unitary, but beamsplitter networks with locality, such as the ones implemented in recent boson sampling and Gaussian boson sampling experiments?

I believe the manuscript makes some nice contributions to the study of entanglement in linear optics and supplies the counterpart to studies of entanglement with Gaussian bosons. However, I believe it can still be improved, especially with regards to writing. I recommend accepting the manuscript if the following (sometimes minor) issues can be fixed.

Requested changes

  • It would be nice to provide details of the numerical implementations, preferably through the sharing of code hosted publicly.

  • I was confused by what the authors mean by the phrase "maximum entropy" in some places. Is it a maximum of the Page curve, meaning the average entropy (averaged over Haar-random unitaries) when subsystem size equals M/2? Or is it truly the maximum achievable Rényi entropy (maximised over all possible unitaries), which is usually larger than the former quantity? This is the analogue of asking whether, for n-qubit systems, the authors are studying the maximum possible entanglement between bipartitions (n/2 log 2) or the typical entanglement at the bipartition size of n/2 (n/2 log 2 - P), where P is the Page correction. This is a crucial point, and while I suspect the authors study the former, the writing can imply in some places that it is the latter. Some examples are on page 13, in the sentence "Thus, in essence, the finding of this last numerical result is that, under collision-free subspace conditions, the maximum entropy that the unitary application can gain is achieved."

  • Page 2, what is meant by "bucket detector"? Is it a detector that cannot resolve photon numbers?

  • I find the second paragraph on page 2 to be too broad an introduction. I don't think understanding the build-up of entanglement "is the holy grail of the field"; I also think the mention of ground states and MPS is completely a distraction.

  • In the same vein, I find the tone of the manuscript a bit boastful when it talks about having an exact expression for the entanglement. It is not a surprise that for linear-optical systems, which are described by quadratic bosonic Hamiltonians, many quantities have analytic expressions. The crucial part is whether these expressions can be efficiently evaluated. Indeed, this is exactly what happens for the output probability of bosonic systems, for which we know a closed-form expression that is unfortunately hard to evaluate.

  • Page 4, the notation |1, 0, 1, . . . , 1⟩, gives the impression that the modes in the ellipses all have a boson number of 1.

  • Page 4, "Permanent calculation is one of the prime examples in the field of computational complexity": a prime example of what?

  • Page 4, "For an n × n matrix, the best-known scaling of the numerical effort for its calculation is of O(n² 2ⁿ⁻¹)": this is a wrong statement because, as the authors point out, there are better known algorithms. So the previously mentioned one cannot have the "best-known scaling".

  • Page 5, typo in "coloumn" vector.

  • Page 5, absent reference in "To generate the numerical results presented in Sec.,"

  • Page 6: are the sᵢs integers? What domain do the xᵢs belong to? Or are they simply formal variables?

  • On page 7, the authors say "The scaling of the numerical effort in terms of mode number is just polynomial and thus almost irrelevant. Overall, this is in stark contrast to the typically much more demanding factorial scaling, according to (M + S − 1)!/[S!(M − 1)!], of the number of basis functions that would be required in a Fock space calculation". As mentioned, this does not mean much in itself. The number of basis functions is not a useful measure of complexity. As a trivial example, I could choose as my basis function the set of all possible outputs of the linear-optical unitary when plugging in different input Fock states. In this case, I can write my output state is trivially a single basis state, but it hasn't simplified my calculation by any amount.

  • Page 10, incomplete bracket in Eq. (23).

  • Page 12, "it has been proven recently that S1 ≥ S2 ≥ S3 . . . [37]". To the best of my knowledge, this is not proved in [37], but was well-known earlier. Although, reading [37], the authors there do mention why the Rényi entropy is in itself also a nice measure to study (regardless of the relation with the von-Neumann entropy). Continuing, I think a comparison with the results of [37] would be nice. The latter study Page curves of Rényi entropies with Gaussian bosons, so in the limit of small squeezing, their results should be reproducible using the authors' results here.

  • Page 12, awkward English in "They display that".

  • Page 12, "functional form turns from convex to (almost) concave.": Please give more evidence for this statement.

  • Page 12, "The second statement is in complete agreement" : it is unclear here what the second statement is referring to.

  • Page 12, "Interestingly, the maximum of the entropy (at 250) is only slightly dependent on the Rényi index when the system size is very large (not shown).": is it possible that the maxima would converge in the limit of large S and M? Perhaps the deviations witnessed are finite-size effects?

  • I would like to add that the fact that "the maximum Renyi entropy saturates as a function of mode number" when in the collision-free subspace was known earlier. See, for example, S. Stanisic, N. Linden, A. Montanaro, and P. S. Turner, Generating Entanglement with Linear Optics, Physical Review A 96, 043861 (2017). In the last row of Table I, when looking at equal-sized bipartitions, one finds that even if the number of modes M→∞, the upper bound on the entanglement is determined by n, the number of bosons.

  • Page 20, "A generalized formula for the permanent has been found along similar lines [51, 52]." A pertinent reference might also be U. Chabaud, A. Deshpande, and S. Mehraban, Quantum-Inspired Permanent Identities, Quantum 6, 877 (2022).

---

## Round 2 · Referee Report · Anonymous (Referee 2) · 2023-4-26

Report

I have seen the changes the authors have made and think they have improved the manuscript. I believe it can be accepted in the journal.

---

## Round 2 · Author Response

Dear Editor-in-charge,

Thank you very much for gathering the very helpful referee reports on our manuscript. In the following, please find a list of corresponding changes that we made in order to improve the manuscript, which, herewith, we would like to resubmit.

1st referee: We thank the referee very much for the careful reading of the ms and for pointing out open questions/remarks that we have taken care of as follows:

-regarding the questions raised at the beginning of the report:

One more comment I have for the authors is to study this timescale in more detail: is it a system-size independent time scale? Does it depend strongly on the number of modes? Is there a phase transition associated with this behaviour? And is the behaviour different if one considers not a Haar-random unitary, but beamsplitter networks with locality, such as the ones implemented in recent boson sampling and Gaussian boson sampling experiments?

We thank the referee for these very interesting questions. To keep the focus of our submitted manuscript tight, we have included some of those question in the outlook section (last paragraph):

"In future work, the time-scale at which the entropy reaches its maximum shall be investigated in more detail, e.g., with respect to its dependence on system size and/or mode number"

"....Gaussian boson sampling [40,52,53], might be worthwhile, with a special focus on the time-scale mentioned above."

Another generalization of the present investigations would be to use out-of-time-order-correlation-functions to understand the scrambling of the quantum information with time. This is approach will be followed in future work.

-regarding the itemized list in the report of the first referee:

1) it It would be nice to provide details of the numerical implementations, preferably through the sharing of code hosted publicly.

Unfortunately, the code used to create the results and figures in the ms is consisting of many separate entities and as of right now, it is not yet suitable for public distribution. The code will be shared on request, however.

2) it I was confused by what the authors mean by the phrase "maximum entropy" in some places. Is it a maximum of the Page curve, meaning the average entropy (averaged over Haar-random unitaries) when subsystem size equals M/2? Or is it truly the maximum achievable Rényi entropy (maximised over all possible unitaries), which is usually larger than the former quantity? This is the analogue of asking whether, for n-qubit systems, the authors are studying the maximum possible entanglement between bipartitions (n/2 log 2) or the typical entanglement at the bipartition size of n/2 (n/2 log 2 - P), where P is the Page correction. This is a crucial point, and while I suspect the authors study the former, the writing can imply in some places that it is the latter. Some examples are on page 13, in the sentence "Thus, in essence, the finding of this last numerical result is that, under collision-free subspace conditions, the maximum entropy that the unitary application can gain is achieved.

The maximum entropy that is meant throughout the ms is the maximum of entanglement entropy as a function of subsystem size, as explained explicitly by a new footnote (Ref. [45]):

"defined as the maximal value of the entanglement entropy as a function of subsystem size (after averaging)"

What is not meant by maximum entropy is the maximum value of the entropy at any specific realization of the HRU. Furthermore, in Sec. 4 C, although $t<1$, we can still look at different bipartitions and then the maximum of the entropy is not at equipartition.

3) Page 2, what is meant by "bucket detector"? Is it a detector that cannot resolve photon numbers?

A bucket detector indeed does not count photons but just gives a yes or no answer, if any photon has or has not arrived at the detector, respectively; this is now clarified by a reference to Ref. [2] and new Ref. [5] from the list of references.

4) I find the second paragraph on page 2 to be too broad an introduction. I don't think understanding the build-up of entanglement "is the holy grail of the field"; I also think the mention of ground states and MPS is completely a distraction.

We have deleted the formulation "holy grail of the field" and have replaced it by the milder and more to the point formulation "is in the central focus of the field".

With respect to the mentioning of MPS: we talk about this method because it is also applied to boson sampling (see Refs. [11,13]) and because of its use in the solid state physics context, where it is employed to describe entanglement dynamics in many-body systems. We have shortened our discussion of MPS by removing the mention of 1D gapped systems (which might be distractive) and are now writing:

"In lattice systems relevant for solid-state physics, e.g., matrix product state (MPS) calculations are mainly favorable for area law scaling of the entanglement growth [20]."

5) In the same vein, I find the tone of the manuscript a bit boastful when it talks about having an exact expression for the entanglement. It is not a surprise that for linear-optical systems, which are described by quadratic bosonic Hamiltonians, many quantities have analytic expressions. The crucial part is whether these expressions can be efficiently evaluated. Indeed, this is exactly what happens for the output probability of bosonic systems, for which we know a closed-form expression that is unfortunately hard to evaluate.

We thank the referee for this comment and have tried to use a more modest language (see answer to previous item, as well the change of the wording "stark contrast" to "clear contrast" in the paragraph after Eq. 13).

6) Page 4, the notation 1, 0, 1, . . . , 1⟩, gives the impression that the modes in the ellipses all have a boson number of 1.

We thank the referee very much for pointing out this inconsistency and have explicitly added some additional zeros in the Fock state now, in order to avoid any misunderstanding.

7) Page 4, "Permanent calculation is one of the prime examples in the field of computational complexity": a prime example of what?

We thank the referee very much for pointing out this omission and have added the wording "...prime example of #P-hard problems..."

8) For an n × n matrix, the best-known scaling of the numerical effort for its calculation is of ${\mathcal O}(n^22^{n-1}$): this is a wrong statement because, as the authors point out, there are better known algorithms. So the previously mentioned one cannot have the "best-known scaling".

The discussion of scaling has now been made more precise by saying "...the scaling of the numerical effort for its calculation via Ryser's formula [27] is of ${\mathcal O}(n^22^{n})$, or ${\mathcal O}(n2^{n})$ using Gray code [28].." and including a new, recent reference (Ref. [28]) on the topic.

9) Page 5, typo in "coloumn" vector}

Typo is corrected.

10) Page 5, absent reference in "To generate the numerical results presented in Sec.,"

Section reference is corrected.

11) Page 6: are the $s_i$s integers? What domain do the $x_i$s belong to? Or are they simply formal variables?

We thank the referee for pointing out the missing definition of the $s_i$. This has now been clarified by including the wording: "... with integers $s_i\geq 0$.." right after Eq. 9. Also it is now made clear explicitly, that the $x_i$ are just formal variables.

12) On page 7, the authors say "The scaling of the numerical effort in terms of mode number is just polynomial and thus almost irrelevant. Overall, this is in stark contrast to the typically much more demanding factorial scaling, according to [(M+S-1)!/S!(M-1)!], of the number of basis functions that would be required in a Fock space calculation". As mentioned, this does not mean much in itself. The number of basis functions is not a useful measure of complexity. As a trivial example, I could choose as my basis function the set of all possible outputs of the linear-optical unitary when plugging in different input Fock states. In this case, I can write my output state is trivially a single basis state, but it hasn't simplified my calculation by any amount.

We thank the referee for pointing this out for clarity. We meant we can handle a large number of modes without much additional numerical costs for a given number of particles. We agree with the referee that the number of basis functions would not be a useful measure of complexity. Still, we consider it a critical condition for efficient simulation and exploited it for our numerical simulation. We modified a sentence in the manuscript for clarity:

"The scaling of the numerical effort in terms of mode number is just polynomial and thus almost irrelevant." replaced by

"The numerical overhead in terms of mode number scales polynomially for a given number of particles, and thus we can handle a larger number of modes efficiently."

13) Page 10, incomplete bracket in Eq. (23).

Unmatched bracket removed

14) Page 12, "it has been proven recently that $S_1\geq S_2 \geq S_3..$ [37]". To the best of my knowledge, this is not proved in [37], but was well-known earlier. Although, reading [37], the authors there do mention why the Rényi entropy is in itself also a nice measure to study (regardless of the relation with the von-Neumann entropy). Continuing, I think a comparison with the results of [37] would be nice. The latter study Page curves of Rényi entropies with Gaussian bosons, so in the limit of small squeezing, their results should be reproducible using the authors' results here.}

We thank the referee for pointing out the fact that the inequality $S_1\geq S_2 \geq S_3..$ is quite well-known. This now reflected by the statement

"...is well-known that $S_1\geq S_2 \geq S_3..$ [39]; see also [40] for a recent discussion of R{\'e}nyi entropy inequalities in the context of Gaussian boson sampling." with a reference to the text-book by Beck and Schlögl (previous Ref [41], now Ref. [39]).

With respect to a comparison of results in new Ref. [40] with ours, we found that we cannot just take the limit of small squeezing parameters but would rather have to use standard Glauber coherent states as basis functions instead of the generalized coherent states we use herein. This could be the topic of future investigations.

15) Page 12, awkward English in "They display that".

We replaced "They display that.." by "They show that..."

16) Page 12, "functional form turns from concave to (almost) convex.": Please give more evidence for this statement.

In parenthesis, we added a supportive statement "...from concave to (almost) convex (the second order derivative (assuming the abscissa to be a continuous variable) of the blue curve in Fig.\ 2 is negative, whereas the second derivative of the green curve is almost everywhere positive)...." and also exchanged the wording "convex" and "concave" right before (which had been in the wrong order)

17) Page 12, "The second statement is in complete agreement" : it is unclear here what the second statement is referring to.

We thank the referee very much for pointing out this unclear formulation. The sentence in question has been removed and a statement in parenthesis has been added at the appropriate position (now page 13):

"..the entropy is monotonically decreasing (in complete agreement with classical results from symbolic dynamics [39]),.."

18) Page 12, "Interestingly, the maximum of the entropy (at 250) is only slightly dependent on the Rényi index when the system size is very large (not shown).": is it possible that the maxima would converge in the limit of large S and M? Perhaps the deviations witnessed are finite-size effects?

To shed some more light on this question, we add the following sentence on page 13:

"Interestingly, the maximum of the entropy (at $M=250$) for $S=10$ is only slightly dependent on the R{\'e}nyi index but we stress that the deviations of the maximum are not finite size effects, because, for different index the maximum entropy will increase linearly with the particle number (with an decreasing slope for increasing $\alpha$) , when the mode number is very large (not shown)."

19) I would like to add that the fact that "the maximum Renyi entropy saturates as a function of mode number" when in the collision-free subspace was known earlier. See, for example, S. Stanisic, N. Linden, A. Montanaro, and P. S. Turner, Generating Entanglement with Linear Optics, Physical Review A 96, 043861 (2017). In the last row of Table I, when looking at equal-sized bipartitions, one finds that even if the number of modes $M\to\infty$, the upper bound on the entanglement is determined by n, the number of bosons.

We thank the referee for pointing out the very important reference, which we now have included at the appropriate position (new Ref. [48]), together with the following wording on page 14:

"This scaling is linear, corroborating the finding displayed in the last entry of Table I in [48], but in contrast to the observation of a logarithmic scaling in [13] for a nonlinear optical network."

In addition, we have changed the wording in the conclusions accordingly: "....could corroborate that under collision-free subspace conditions, the maximum R{\'e}nyi entropy saturates as a function of mode number."

20) Page 20, "A generalized formula for the permanent has been found along similar lines [52, 53]." A pertinent reference might also be U. Chabaud, A. Deshpande, and S. Mehraban, Quantum-Inspired Permanent Identities, Quantum 6, 877 (2022).

We thank the referee for pointing out the very helpful reference by Chabaud et al, which is now included at the end of the first appendix by the sentence:

"Furthermore, it is worthwhile to note that different permanent identities have been proven in a quantum inspired way in [58]. The Glynn formula has, e.g., been proven using cat states."

2nd referee: We thank the referee very much for the careful reading of the ms and for pointing out open questions/remarks that we have taken care of as follows:

1) experiments performed already on Boson sampling etc.

A citation of a review of BS experiments (new Ref. [5]) is now included in the first paragraph, the end of which reads:

"A review of the many different variants of boson sampling and their experimental realization as well as its validation, including references to pioneering studies, is given in [5]".

2) experimental possibilities in implementing the research studied here in detail.

A discussion of a possible measurement scenario for the experimental determination of Renyi entropies, including new Ref. [43] has been added:

"An experimental approach to measuring R{\'e}nyi entropies employs the preparation of two copies of the same system and measuring the expectation value of the so-called swap operator. It has originally been devised to investigate quench dynamics in Bose-Hubbard type lattice Hamiltonians by Daley et al. [43]."

On behalf of all coauthors. Sincerely yours, Yulong Qiao

---

## Round 2 · List of Changes

0) We have introduced numbered sections that can be referenced to

1) page 2, first paragraph: inclusion of word “simple” such that sentence reads: “....simple bucket detector.....”

2) page 2, at end of first paragraph: “A review of the many different variants of boson sampling and their experimental realization as well as its validation, including references to pioneering studies, is given in [5]”.

3) page 2, second paragraph: We have deleted the formulation “holy grail of the field” and have replaced it by the milder and more to the point formulation “is in the central focus of the field”.

4) page2/page3: instead of “In matrix product state (MPS) calculations, this growth is either determined by an area law in a ground state 1D gapped system [20] or by a more intractable scaling in other cases.” we are now writing: “In lattice systems relevant for solid-state physics, e.g., matrix product state (MPS) calculations are mainly favorable for area law scaling of the entanglement growth [20].”

5) page 4: we replaced |1,0,1,\dots,1\rangle by |1,0,1,0\dots,0,1\rangle

6) bottom of page 4: we have added the wording “...prime example of #P-hard problems...”

7) bottom of page 4: reformulation of sentence ``For an $n\times n$ matrix, the best-known scaling of the numerical effort for its calculation is of ${\mathcal O}(n^22^{n-1})$ [..], or ${\mathcal O}(n2^{n-1})$ using Gray code, as compared to ${\mathcal O}(n^{2.373})$ for the determinant.''

to

``....the scaling of the numerical effort for its calculation via Ryser's formula [27] is of ${\mathcal O}(n^22^{n})$, or ${\mathcal O}(n2^{n})$ using Gray code [28]....''

8) page 5 after Eq. (5): correction of type error “coloumn” replaced by “column”

9) page 5: inclusion of Section reference Section reference is corrected (see also item 0).

10) page 6 now includes the wording: “... with integers si ≥ 0..” right after Eq. (9). and “....formal variables...” on the last line.

11) page 7: the sentence “The scaling of the numerical effort in terms of mode number is just polynomial and thus almost irrelevant.” is replaced by “The numerical overhead in terms of mode number scales polynomially for a given number of particles, and thus we can handle a larger number of modes efficiently.”

12) page 8: replacement of “stark” by “clear”

13) page 10: unmatched bracket in Eq. (23) removed

14) page 12: replacement of “...and it has been proven recently that S1 ≥ S2 ≥ S3 . . . [..].” by “...is well-known that S1 ≥ S2 ≥ S3.. [39]; see also [40] for a recent discussion of R´enyi entropy inequalities in the context of Gaussian boson sampling.”

15) page 12: addition of footnote [45]: “defined as the maximal value of the entanglement entropy as a function of subsystem size (after averaging)”

16) page 12: addition of sentences: “An experimental approach to measuring R´enyi entropies employs the preparation of two copies of the same system and measuring the expectation value of the so called swap operator. It has originally been devised to investigate quench dynamics in Bose-Hubbard type lattice Hamiltonians by Daley et al. [43].”

17) (new) page 13, replacement of “They display that..” by “They show that...”

18) (new) page 13: addition of a statement in parenthesis “..the entropy is monotonically decreasing (in complete agreement with classical results from symbolic dynamics [39]),..”

19) (new) page 13: in parenthesis, we added a supportive statement “...from concave to (almost) convex (the second order derivative (assuming the abscissa to be a continuous variable) of the blue curve in Fig. 2 is negative, whereas the second derivative of the green curve is almost everywhere positive).” and also exchanged the wording “convex” and “concave” right before (which had been in the wrong order)

20) page 13, addition of new sentence: “Interestingly, the maximum of the entropy (at M = 250) for S = 10 is only slightly dependent on the R´enyi index but we stress that the deviations of the maximum are not finite size effects, because, for different index the maximum entropy will increase linearly with the particle number (with an decreasing slope for increasing α) , when the mode number is very large (not shown).”

21) (new) page 14: replacement of sentences “The scaling of the maximum entropy with respect to particle number can be read in Figure 1. It seems that also this scaling is linear, in contrast to the observation of a logarithmic scaling in [..] for a nonlinear optical network.” by “The scaling of the maximum entropy with respect to particle number can be extracted from Figure 1. This scaling is linear, corroborating the finding displayed in the last entry of Table I in [48], but in contrast to the observation of a logarithmic scaling in [13] for a nonlinear optical network.”

22) page 19: change of wording from “Furthermore, because of the polynomial dependence on M of our numerical complexity, we could uncover that...” to “....could corroborate that under collision-free subspace conditions, the maximum R´enyi entropy saturates as a function of mode number.”

23) page 19: inclusion of two statements in last paragraph of conclusion: “In future work, the time-scale at which the entropy reaches its maximum shall be investigated in more detail, e.g., with respect to its dependence on system size and/or mode number” “....Gaussian boson sampling [40,52,53], might be worthwhile, with a special focus on the time-scale mentioned above.”

24) page 20: inclusion of new sentence: “Furthermore, it is worthwhile to note that different permanent identities have been proven in a quantum inspired way in [58]. The Glynn formula has, e. g., been proven using cat states.”

---

## Editorial Decision

published